# Bayesian active sound localisation: To what extent do humans perform like an ideal-observer?

Glen McLachlan[1]*, Piotr Majdak[2], Jonas Reijniers[1], Michael Mihocic[2], Herbert Peremans[1]

**1** Department of Engineering Management, University of Antwerp, Antwerp, Belgium, **2** Acoustics Research Institute, Austrian Academy of Sciences, Vienna, Austria

* glen.mclachlan@uantwerpen.be

**Data Availability Statement:** All behavioural data and MATLAB code for the model and its output are available under the name mclachlan2025, as part

## Abstract

Self-motion is an essential but often overlooked component of sound localisation. As the directional information of a source is implicitly contained in head-centred acoustic cues, that acoustic input needs to be continuously combined with sensorimotor information about the head orientation in order to decode to a world-centred frame of reference. When utilised, head movements significantly reduce ambiguities in the directional information provided by the incoming sound. In this work, we model human active sound localisation (considering small head rotations) as an ideal observer. In the evaluation, we compared human performance obtained in a free-field active localisation experiment with the predictions of a Bayesian model. Model noise parameters were set a-priori based on behavioural results from other studies, i.e., without any post-hoc parameter fitting to behavioural results. The model predictions showed a general agreement with actual human performance. However, a spatial analysis revealed that the ideal observer was not able to predict localisation behaviour for each source direction. A more detailed investigation into the effects of various model parameters indicated that uncertainty on head orientation significantly contributed to the observed differences. Yet, the biases and spatial distribution of the human responses remained partially unexplained by the presented ideal observer model, suggesting that human sound localisation is sub-optimal.

## Author summary

By moving our heads, we can obtain additional information about the direction of a sound. This requires the integration of acoustic and sensorimotor information. To understand this process better, we formulated an ideal observer model for active sound localisation, which provided the Bayesian optimal response, given the available information. We then compared the model's predictions to the results from a behavioural localisation experiment with sources presented from loudspeakers at a wide distribution of directions. While the model generally matched human performance, it could not accurately predict the bias and spread of localisation estimates for stimuli from certain directions, most

of the Auditory Modeling Toolbox (AMT) at link https://amtoolbox.org/.

**Funding:** This research was supported by the Research Foundation Flanders (FWO, under grant number G023619N to HP), the Agency for Innovation and Entrepreneurship (VLAIO, grant number HBC.2019.2191 to JR), and the European Union (EU, project "SONICOM", RIA action of Horizon 2020, under grant number 101017743 to PM). The funders had no role in study design, data collection and analysis, decision to publish, or preparation of the manuscript. JR received a salary from grant no. HBC.2019.2191 and grant no. G023619N, GM received salary from grant no. G023619N.

**Competing interests:** The authors have declared that no competing interests exist.

notably from above and behind the listener. We found that uncertainty about head position played a significant role in this discrepancy. Still, the distributions of human responses were not fully explained by our model, suggesting that humans may utilise the information available to them sub-optimally.

## Introduction

The acoustic cues acquired through head rotation are of crucial importance to spatial hearing. Not only do they improve sound externalisation [1], they provide dynamic acoustic cues which contribute to sound localisation. First and foremost, they reduce ambiguities between the front and back [2, 3]. Second, under certain conditions, they can improve the elevation estimation of a source through a relation between the rate of change in binaural cues and the amount of performed head rotation, a phenomenon termed the 'Wallach cue' [2, 4]. Dynamic acoustic cues become especially important when spectral cues are difficult to process, like in reverberant [5] or virtual [6] environments, or when high-frequency cues are unavailable [7].

The movements that benefit sound localisation are not always voluntary. On the contrary, for both sensorimotor [8] and behavioural [9, 10] reasons, the human head is rarely completely still. Even when the only task given to subjects is to remain still, they continue to move by a small but measurable amount [11]. Despite this constant motion, we perceive the auditory world to be relatively stable. This suggests that there exists a mechanism that utilises positional information about the head to compensate for self-motion and converts the head-centred auditory cues into a stable, world-centred frame of reference [12, 13]. This notion is further supported by the fact that moving sound sources do not provide the same benefits to localisation as the head movements that would theoretically cause similar acoustic cues [14].

If we are to include these unavoidable dynamic effects in future studies of sound localisation, it is apparent that there is a need for a tool to better investigate or predict the effects of head movements in a reproducible manner. To this end, we previously proposed an ideal observer model for active sound localisation based on Bayesian inference, which can process dynamic cues obtained through self-motion. This model integrates acoustic and sensorimotor information over time to simulate sound localisation through self-motion [15]. This model serves as a performance 'ceiling', given the available acoustic cues. It also provides a bottom-up approach to sound localisation, i.e., it lets one change the cues that are extracted from incoming sound and the way that they are utilised (e.g. by adjusting the spatial prior or increasing sensory noise), after which the effects on localisation performance can be tested.

In this paper, we investigated to what extent humans behave like an ideal observer during active sound localisation. We did this by comparing the output of a Bayesian model for active sound localisation to behavioural data over the full 2D sphere. First, we described the active sound localisation model. This model continually collects auditory snapshots or 'looks', akin to the multiple looks model [16]. Through recursive Bayesian estimation, these looks were accumulated over time, reducing sensory ambiguity. The use of snapshots means that dynamic cues were formed implicitly, i.e., the additional information obtained from head motion was obtained through a series of static looks. Note that this model controls head movement irrespective of the incoming sound and, hence, does not encompass 'closed-loop' processes such as triangulation or source tracking. Next, we described the localisation experiment and compared the results obtained here to the model data. Finally, we adjusted a subset of the parameters to investigate their effect on localisation performance. The present experiments focus on small head rotations (10°) along the yaw axis. Small head movements have been shown to

comprise the majority of natural head movements [17], and can be considered, if necessary, as a first step in a more complex movement framework. All experimental data, including the localisation model, were made publicly available in the Auditory Modeling Toolbox (AMT) [18].

## Methods

### Ethics statement

All subjects that participated in this study were adult volunteers. They were informed on the procedure and were free to withdraw at any time. They gave written informed consent before the experiment. The study applied the standard methodology of the Acoustics Research Institute (ARI) which has been approved by the Ethics Representatives of the ARI.

### Template definition

The proposed model utilises a 'template-matching' procedure which requires a set of acoustic templates $\mathbf{T}_A$ that the observed information is compared to [19–21]. Each template in $\mathbf{T}_A$ contains the expected acoustic information from a specific direction. We assumed that $\mathbf{T}_A$ is the acoustic 'knowledge' that the brain has learned and stored over a lifetime of experience, and is thus signal-independent.

To compute $\mathbf{T}_A$, a set of acoustic features was extracted from the subject's head-related transfer functions (HRTFs) and the signals received at each ear. This process is identical to the feature extraction described in earlier work [21].

The ITD template $T_{itd}$ was computed as the difference between times of arrival (TOAs) of the head-related impulse responses (HRIRs) at each ear. The TOA was defined as the time it takes for the HRIR to reach a value 10 dB below its maxima. Each HRIR was low-pass filtered at a cutoff frequency of 3000 Hz before deriving the TOA. Then, the ITDs (in time units) were transformed into a scale of just-noticeable difference (JND) units, such that the error on the ITD was modeled as an additive instead of multiplicative factor (for further explanation, see [21]).

Next, we consider the directional filters for the left $\mathbf{T}_L$ and right $\mathbf{T}_R$ ear separately. The HRIRs were passed through a Gammatone filterbank with 32 channels in equivalent rectangular bandwidths (ERBs), with centre frequencies ranging between 300 Hz and 15 kHz, as in [21]. These processed signals were half-wave rectified, low-pass filtered using five sequential first-order infinite impulse response (IIR) filters with a cut-off frequency of 2000 Hz, and then transformed to a logarithmic domain (in dB). This stage simulates a simplified processing of the inner hair cell [22]. Then, for each frequency channel, the root mean square of the signal was computed. Thus, $\mathbf{T}_L$ and $\mathbf{T}_R$ denote vectors with monaural spectral information in dB along the ERB channels.

Ultimately, the ITD and the monaural spectral vectors for both ears are combined into $\mathbf{T}_A$ which is a matrix containing the combined vectors per template source direction:

$$\mathbf{T}_A = [T_{itd}, \mathbf{T}_L, \mathbf{T}_R] \tag{1}$$

In this article, $\mathbf{T}_A$ consists of 2042 directions that are uniformly distributed over the sphere. These directions were obtained through spherical-harmonics interpolation of the measured HRTFs, involving Tikhonov regularisation to account for measured directions not covering the full sphere [23].

## Generative model

We assume that the listener wants to determine the source direction based on all prior information about the environment and all sensory information collected during the head movement. This prior and sensory information is combined into a posterior probability density function (PDF), from which finally a point estimate is retrieved.

To explain this process step-by-step, we first introduce the function of the spatial prior $p(\psi)$. Then we discuss how the likelihoods of the acoustic information $L_A$ and sensorimotor information $L_H$ are computed. Finally we combine these factors into the posterior PDF and obtain an estimate of the sound source direction from this distribution.

**Spatial prior.**   In the Bayesian framework the probability of an occurring event may be affected by prior knowledge about the event. The spatial prior $p(\psi)$ quantifies the listener's a-priori assumptions about the source location before taking any sensory information into account. Polar estimations show a general bias towards the audio-visual horizon [24, 25]. This can be modelled with a Gaussian spatial prior around the horizontal plane with a limited SD of about 12˚. However, the best fitting SD of the prior seems to depend on the decision rule used.

The spatial prior is only one example of possible prior information available to a listener. Priors can be related to any variable, such as the number of sources [26], the movement properties of the sound source [27] or its spectral content [28]. In fact, the proposed model relies on the assumption that the source spectrum is unknown, but is derived from an ecologically valid prior.

**Acoustic sensor model.**   The acoustic sensor model compares the stored template information $\mathbf{T}_A$ to a vector of acoustic features present in the observed sound signal, $\mathbf{y}_A$, which consists of the noiseless 'true' state of the acoustic information, $\mathbf{X}_A$, corrupted with noise due to uncertainties within the auditory system or caused by the environment:

$$\mathbf{y}_A = [y_{itd}, \mathbf{y}_L, \mathbf{y}_R] \tag{2}$$

with

$$y_{itd} = X_{itd} + \delta_{itd}$$
$$\mathbf{y}_L = \mathbf{X}_L - \hat{\mathbf{S}} + \delta_L + \delta_S$$
$$\mathbf{y}_R = \mathbf{X}_R - \hat{\mathbf{S}} + \delta_R + \delta_S$$

where $\delta_{itd}$ is the error on the ITD measurement with standard deviation $\sigma_{itd}$, $\delta_L$ and $\delta_R$ are the errors on the left and right monaural spectra measurements with covariance matrices $\mathbf{\Sigma}_L = \mathbf{\Sigma}_R = \sigma_I^2 \cdot \mathbf{I}$, respectively. Thus, $\sigma_I$ represents the noise on the spectral measurements. Finally, $\hat{\mathbf{S}}$ is the mean expected source spectrum and $\delta_S$ is the error due to imperfect knowledge of the sound source with covariance matrix $\Sigma_S$ (assuming a central process, this is the same at both ears). So, $\hat{S}$ and $\Sigma_S$ define the observer's prior on the source spectrum:

$$P(S) = \mathcal{N}(\hat{S}, \Sigma_S) \tag{3}$$

i.e., the observer assumes that the source has spectrum $\hat{S}$ with an uncertainty which is contained by the source covariance matrix $\Sigma_S$.

With that, we define the full covariance matrix of the acoustic cues as:

$$\mathbf{\Sigma}_A = \begin{bmatrix} \sigma_{itd}^2 & 0 & 0 \\ 0 & \mathbf{\Sigma}_L + \mathbf{\Sigma}_S & \mathbf{\Sigma}_S \\ 0 & \mathbf{\Sigma}_S & \mathbf{\Sigma}_R + \mathbf{\Sigma}_S \end{bmatrix} \tag{4}$$

All error terms are assumed to be zero-mean Gaussian noise. Note that we assume the spectrum to be time-invariant and the source in the far field. We also assume here that each frequency channel has the same sensory noise of $\sigma_I$. Several studies have shown that the JND shows little dependence of signal frequency for sources louder than 50 dBA SPL [29, 30].

We then consider the acoustic sensor model:

$$L_A(t_i) = p(\mathbf{y}_A(t_i) \mid \theta_H(t_i), \psi) \tag{5}$$

where $\mathbf{y}_A(t_i)$ and $\theta_H(t_i)$ are the observed acoustic information and the true head orientation, respectively, at time-step $t_i$, and $\psi$ is the true sound source direction, which here is assumed to be independent of time.

The expression in Eq 5 is calculated by computing the Mahalanobis distance between the measured acoustic cues $\mathbf{y}_A(t_i)$ and the set of acoustic cue templates $\mathbf{T}_A$ and covariance matrix $\Sigma_A$. This is done at each sampled sound source direction, given the current head orientation $\theta_H(t_i)$.

**Motor sensor model.** The motor sensor model is defined as:

$$L_H(t_i) = p(\theta_H(t_i) \mid y_H(t_0 : t_i), u(t_0 : t_i)) \tag{6}$$

where $\theta_H(t_i)$ and $y_H(t_i)$ are the true and observed head orientations at each time step $t_i$, respectively, and $u$ is the motor command signal, which is represented by the speed $\omega(t_i)$ of rotating the head around a given axis. These variables are defined as:

$$\begin{aligned} y_H(t_i) &= \theta_H(t_i) + \delta_H, \\ \theta_H(t_{i+1}) &= \theta_H(t_i) + u(t_i)\Delta t + \delta_u, \end{aligned} \tag{7}$$

The additive noise on both the movement equation and the sensor equation is again assumed to be zero-mean white Gaussian noise $\delta_u \sim \mathcal{N}(0, \sigma_u)$ and $\delta_H \sim \mathcal{N}(0, \sigma_H)$. Thus, $\sigma_H$ describes the noise on the head orientation observation at each time step and $\sigma_u$ describes the noise on the motor command that steers the head.

Assuming head orientation measurements to be independent of acoustic measurements, we show in [15] that Eq 6 can be reformulated. The dependency on all sensor readings and all head rotations executed so far can be expressed recursively as

$$\begin{aligned} L_H(t_i) &= p(\theta_H(t_i) \mid y_H(t_0 : t_i), u(t_0 : t_i)) \\ &= p(\theta_H(t_i) \mid \hat{\theta}_H(t_i)) \end{aligned} \tag{8}$$

with $\hat{\theta}_H(t_i) \sim \mathcal{N}(\mu_{\theta_H}(t_i), \sigma_{\theta_H}(t_i))$ the estimated head orientation updated at each step through a Kalman filter with:

$$\begin{aligned} \mu_{\theta_H}(t_{i+1}) &= (1 - K) \cdot (\mu_{\theta_H}(t_i) + u(t_i)\Delta t) + K \cdot y_H(t_{i+1}), \\ \sigma_{\theta_H}^2(t_{i+1}) &= (1 - K) \cdot (\sigma_{\theta_H}^2(t_i) + \sigma_u^2) \end{aligned}$$

and K the Kalman gain:

$$K = \frac{\sigma_{\theta_H}^2(t_i) + \sigma_u^2}{\sigma_{\theta_H}^2(t_i) + \sigma_u^2 + \sigma_H^2},$$

(9)

The expression in Eq 8 is calculated by computing the Mahalanobis distance between given head orientation $\theta_H(t_i)$ and $\hat{\theta}_H(t_i)$.

At the initial time step $t_0$ we define: $\mu_{\theta_H}(t_0) = y_H(t_0)$ and $\sigma_{\theta_H}^2(t_0) = \sigma_H^2$.

**Posterior computation.** By marginalisation over all possible head orientations and using Bayes' theorem, we can combine the spatial prior and sensor model output to obtain the joint posterior PDF:

$$p_{t_i} = C \cdot p_{t_{i-1}} \times \int_{\theta_H} (L_H \times L_A) d\theta_H,$$

(10)

Turning to Bayesian terminology, $p_{t_i}$ is the posterior PDF, $p_{t_{i-1}}$ is the prior PDF and the joint sensor model computes the likelihood. $C$ is a normalisation constant. Note that the prior at time step $t_i$ equals the posterior from time step $t_{i-1}$. At the initiation of the cumulative process, $p_{t_{i-1}} = p(\psi)$, which is the spatial prior. The detailed derivation of this equation is explained in [15].

Fig 1 illustrates how the posterior updates over time as more information arrives. Note that in the numerical implementation of the model, the initial look contains all acoustic information, and the following looks only consider changes in the ITD cue to update the posterior. The reasoning for this is explained in the Methods.

**Localisation decision rule.** Eq 10 returns a probability distribution over the sphere, i.e., a probability from a large but discrete set of source directions. The last step in the process is to obtain a point estimate from this posterior PDF. To do so, a decision rule must be defined. The present model uses the posterior matching (PM) strategy, where a weighted random sample is taken from the posterior PDF. However, it is easy to implement other strategies, such as the maximum a posteriori (MAP) strategy, i.e., selecting the location at the maximum of the posterior. It was found that localisation performance lies somewhere between the MAP and PM strategies [24].

## Model parameters

**General.** The stimulus was a broadband time-invariant Gaussian white noise burst. The duration was the same as in the behavioural experiment, for both movement conditions. The simulated head movement was copied directly from the experimental head tracker data. In other words, for each simulated trial, the model executed the same head rotation as the subject did during the experiment.

The template $\mathbf{T_A}$ was listener-specific, i.e., derived from the individual's measured HRTFs. Earlier work suggests that the auditory system can detect changes (ITD [31], ILD [31, 32], spectrum [28, 33]) on a short time scale of about 5ms. However, the full integration window of acoustic information for sound localisation appears to be more in the range of 100-200ms [34, 35]. Furthermore, in localisation studies along the horizontal plane, the azimuth estimation reached best performance for stimulus durations of only 3 ms [36]. In studies along the vertical plane, a longer duration of 80 ms was required to reach the best performance in the elevation estimation [28]. When head movements are allowed, a stimulus duration of approximately 100 ms seems to be required to provide a substantial benefit from the head movement [37]. For the

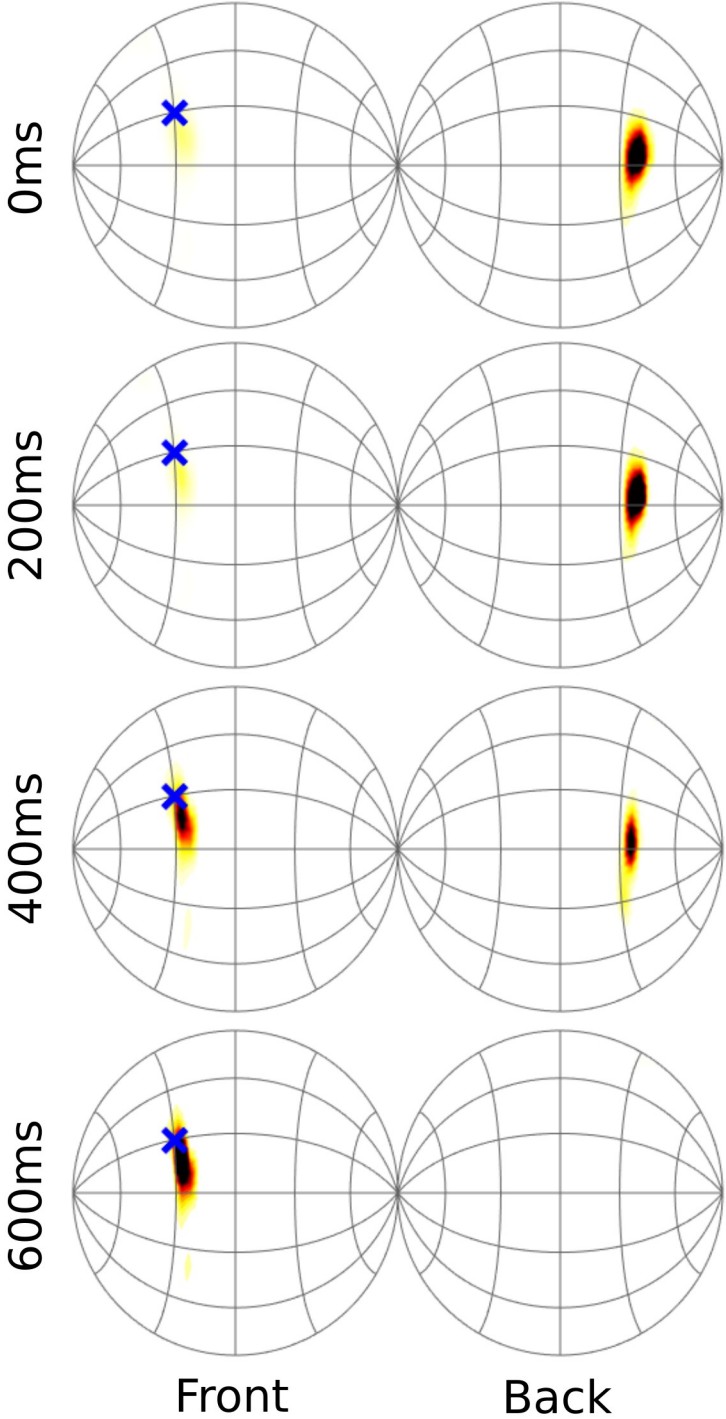

**Fig 1. Example posterior distribution of source direction at different time steps during yaw rotation.** Darker areas indicate higher probabilities. The blue 'x' is the true source direction.

above reasons, the time step size $\Delta t$ for the updating of the posterior was set to 100ms. The effects of other step sizes are reported in the discussion.

The localisation task was simulated for 33 source directions and repeated 20 times without head movement, 20 times with a 10˚ rotation to the left, and 20 times to the right. This reflects the sound directions and repetitions used in the behavioral experiment. The simulations were repeated for each subject that participated in the behavioral experiment, then the results were pooled for further statistical analyses.

The spatial prior was assumed to emphasise directions around the horizon [24, 38]. That is, for elevation, the prior has a mean around zero and a restricted variance $\sigma_p$. In [24], the optimal value of $\sigma_p$ was found to be around 11.5˚. Initial simulations in the present model showed that this prior is too strong, as will be shown in the discussion. An explanation may be that this value was determined from a localisation experiment that only included source directions in the elevation range of $[-35˚, 35˚]$ in the frontal hemisphere. For better comparability with the behavioural data, the model spatial prior was weakened to 30˚.

**Acoustic information.** In this implementation of the model, $\mathbf{y_L}$ and $\mathbf{y_R}$ were measured once, and $y_{itd}$ was measured several times during stimulus presentation. The reason for this is twofold. First, it was found that dynamic spectral cues are not informative for sound localisation during small head rotations, which makes it unnecessary to take several measurements of the spectral information during head rotation [6]. Second, the model's recursive estimation process relies on the assumption that the measurements are independent and identically distributed. This assumption does not hold for natural source spectra: we found the spectra of two subsequent segments of 100ms for sources from the ESC-50 database [39] to be highly correlated ($\rho \approx 0.8$).

The SD on the ITD measurement at each time step $t$, $\sigma_{itd}$, was set to 0.6 JND. The SD on the measurement of the spectral content, $\sigma_I$, was set to 3.5 dB. These values and units were derived earlier for the static localisation model in [21]. In the discussion, the effects of higher ITD measurement noise are reported.

The covariance matrix of the knowledge of the spectral content of the incoming sound source, $\Sigma_S$, i.e., the source prior, was derived from the ESC-50 database [39], which is a collection of 2000 environmental audio recordings. Each of the sound files of the database were chopped up in intervals of 0.2s and, for each of these intervals, the source log-magnitude spectrum was expressed as function of the ERB centre frequencies. The resulting spectra were pooled in a single dataset, from which the average source spectrum $\hat{S}$ and the covariance matrix $\Sigma_S$ were calculated.

**Sensorimotor information.** The SD on the measurement of the head orientation at each time step $t$, $\sigma_H$, and the SD on the motor command steering the head rotation, $\sigma_u$, were both initially set to 0˚. In other words, this assumed that the listener can perfectly estimate and control the head orientation. Human subjects are able to report motion and orientation perception with very high precision [40, 41]. In a seated position, the standard deviation of head rotation around the starting position (notated in our model as $\sigma_u$) is around 2˚ [11]. In the discussion the effects of higher motor noise are reported.

As with virtually all sensory systems, motor imprecision increases with stimulus magnitude [42, 43], i.e., noise increases with exerted force. However, motor noise ($\sigma_H$ and $\sigma_u$) was assumed here to be additive for simplicity.

Theoretically, Eq 10 requires marginalisation over all possible head rotations. However, with low sensorimotor noise, the probability distribution of head orientations based on the accumulated sensorimotor evidence will be near-zero for most orientations. Hence, a

computational heuristic was used to only consider orientations within a range of $2\sigma_H$ from the true orientation, so that about 95% of the orientations were considered.

## Acoustic measurements and behavioural experiment

**Apparatus.** The behavioural localisation experiment and acoustic HRTF measurements were conducted in a semi-anechoic room with 91 speakers (E301, KEF Inc.) distributed over the sphere within the elevation angles −47˚ to 90˚. A head-mounted display (HMD, Oculus Rift, CV1, Meta Inc.) was used for the visual presentation of the virtual environment: a sphere with grid lines, which serve as anchor points to the subject's orientation in space. Although HMDs can minimally affect localisation performance [44, 45], this decision was made as a higher precision error was found in darkness than when providing spatial information with an HMD [46]. Three infrared cameras were used for the tracking of the listener within the six degrees of freedom.

The experiment was controlled by a computer running a 64-bit Windows 10, equipped with an 8-core, 3.6-GHz CPU (i7-11700KF, Intel Inc.), 16 GB of RAM, and a graphic card with dedicated 8 GB of RAM (GeForce RTX 3070, NVIDIA Inc.). The experiment was controlled by the ExpSuite 1.1 application LocaDyn, version 0.9.7.

The tracking system provided a translation accuracy of below 1 *cm* [47] and a rotation accuracy of below 1˚ (for a similar tracking system, [48]). The position and orientation of the subject's head were recorded for later analyses and to simulate using the model.

**Subjects.** Eight normal-hearing subjects (four female, four male) participated in the experiment. Their absolute hearing thresholds were within the average (±1 standard deviation, SD) of the age-relevant norms [49, 50] within the frequency range from 0.125 to 12.5 kHz. The age range of the subjects was between 22 and 33 years.

**Stimuli.** Stimuli were always played over loudspeakers using vector base amplitude panning (VBAP) [51]. Thirty-three source directions were distributed over the full sphere, at lateral and polar steps of 30˚.

The acoustic stimulus used in this experiment was a wideband (20 to 20000 Hz) white noise burst, gated with a 10-ms cosine ramp. Each trial used the same noise realisation. The stimulus was gated off after 500 ms in the passive condition, and after 10˚ of head rotation for the active condition. For the latter, this means that the stimulus duration depended on the rotation velocity, with mean 634.9 ms and SD 335.7 ms.

Presentation level was measured to be 48 dBA SPL at the ear drum, with a ±2.5 dB level roving range between trials.

**Procedure.** The localisation task procedure was identical to that of [6]. At the start of each trial, the subject kept the head still on the reference orientation at (0˚, 0˚). The stimulus was then played and the subject remained still or initiated rotation depending on the movement condition. At the end of the stimulus, the subject pointed towards their perceived source direction with a hand-tracking device to provide their localisation estimate. No feedback was provided about performance during or after the trials.

In the condition labelled 'passive', the subject was instructed to keep the head still for the duration of the stimulus. For the condition labelled 'active', the subject was instructed to make a single-sided rotation (either to the left or to the right) as soon as they heard the stimulus onset. Half of the trials instructed a leftward rotation, the other half was rightward. The head rotation speed was unrestricted, but was monitored through the tracking system of the VR headset and recorded for analysis.

In total, the passive experiment consisted of 660 trials (33 directions and 20 repetitions) per subject. The active experiment consisted of 1320 trials per subject: 660 for a leftward rotation

and 660 for a rightward rotation. The trials were divided into 6 blocks, with trials and blocks presented in random order. For passive localisation, trials which exceeded 2˚ of movement in any direction were excluded (225 omissions in total). For active localisation, trials that resulted in a total yaw rotation smaller than 7˚ or larger than 13˚, or with a pitch rotation larger than 6˚ were omitted (648 omissions in total).

Subjects were trained before commencing the experiment. The training consisted of 300 trials with the 500ms white noise burst, played from a direction randomly selected from a uniform distribution within the range of available directions. Subjects were not excluded based on their performance at the end of training.

## Localisation metrics

There are many metrics available for sound localisation performance, this makes comparison between localisation studies difficult. It is, however, generally accepted that a distinction needs to be made between two types of errors [25]. The first type is the local error. Here we use the lateral-polar coordinate system [52], $(\theta, \phi)$, where $\theta \in [-90, 90]$ and $\phi \in (-180, 180)$, with $(\theta, \phi) = (0, 0)$ defined as straight ahead. The local error was expressed in root mean-squared error (RMSE) value of the lateral and polar errors. Polar RMSE was only considered in the range of $\pm 30$˚ lateral angle, and estimates in the wrong hemisphere (i.e., front-back and up-down confusions) were excluded from the local errors, following the definition by Middlebrooks [53].

The second type of error is the reversal error, which generally is reported as a percentage, i.e., the rate of reversals in a given set of trials. The first reversal error considered was the quadrant error (QE) rate, which is defined as any polar error larger than 90˚. Additionally, we used the front-back confusion (FBC) rate and the up-down confusion (UDC) rate, which are defined as any response crossing the frontal plane and the horizontal plane, respectively. This is the same definition as used by Carlile et al. [25], and thus allows for a direct comparison between present and previous data. Note that this is a very coarse definition for the reversal error, as it confounds FBCs and UDCs with local errors near the frontal or horizontal plane, respectively.

## Results

### Global statistics

First, we present the global results, in order to compare to the existing literature. Table 1 presents the localisation data of both the behavioural (B) experiment and the model (M) simulations (means and SDs averaged over the subjects), for passive (P) and active (A) conditions.

**Local errors.** The lateral errors in the behavioural results agreed with previous findings from similar experimental setups [53, 54]. However, the model results were small compared to

**Table 1. Averages and SDs of behavioural (B) and modelled (M) localisation performance in the passive (P) and active (A) conditions.** The performance is represented as the lateral and polar RMSE (in degrees), QE, FBC, and UDC rates (in %). Means and SDs were computed over eight (virtual) subjects. For comparison, the results from previous work are reported too [53]. N.R.: not reported.

| Condition | L. RMSE (deg) | P. RMSE (deg) | QE (%) | FBC (%) | UDC (%) |
|---|---|---|---|---|---|
| BP [53] | 10.6 ± 2.0 | 22.7 ± 5.1 | 4.6 ± 5.9 | N.R. | N.R. |
| BP | 8.1 ± 1.4 | 25.1 ± 3.2 | 7.9 ± 4.5 | 10.7 ± 6.3 | 6.7 ± 4.2 |
| BA | 8.2 ± 1.8 | 20.5 ± 5.2 | 0.7 ± 1.1 | 1.5 ± 1.2 | 4.2 ± 4.1 |
| MP | 2.7 ± 0.5 | 21.8 ± 3.4 | 7.4 ± 2.0 | 4.0 ± 1.3 | 3.1 ± 0.5 |
| MA | 2.7 ± 0.2 | 18.4 ± 2.3 | 2.0 ± 1.1 | 1.1 ± 0.3 | 1.8 ± 0.7 |

the behavioural data. This may have been the result of $\sigma_{itd}$ being set too small, or of the behavioural responses being confounded with a 'pointing' error. The effect of the ITD noise on the model performance is tested below.

The polar errors of the behavioural results were consistent with previous findings [6, 53]. Like the lateral error, the mean polar error of the simulated trials was lower than that of the behavioural data. However, the difference here was smaller.

No decrease was seen in polar error in condition BA. This is evidence against the 'Wallach cue' [3], and agrees with previous findings [6]. However, condition MA did show a decrease in polar error. Although this improvement is still small, it is an indicator that the Wallach cue may be theoretically informative, as the model was able to obtain elevation information from yaw movement. The reason why this isn't seen in humans could be due to motor noise. This is investigated below.

Interestingly, the SD of the polar RMSE increased in condition BA, even though the mean did not change much. This suggests that the effects of head movement may be subject-dependent, e.g., motor noise during motion may be higher for some individuals.

**Reversal errors.** The QE rates in conditions BP and MP agreed with previous work [53]. Furthermore, the near-complete removal of QEs in conditions BA and MA also confirms the consensus that head rotation resolves all reversal errors. [6].

The FBC rate in condition MP was notably lower than in condition BP. Looking at earlier studies, FBC rates of normal hearing listeners were closer to 3–6% [25, 55, 56], although the errors were highly subject-dependent. This means that the high FBC rate in this study is somewhat anomalous. Hence, the cause for the discrepancy here seems to lie in the behavioural results, not in the model predictions.

There was also a slight decrease in the UDC rate. This reduction may have been caused by the improved polar estimation obtained from the Wallach cue. It is possible that the rotation made during some trials contained a significant roll-component, which helps distinguish between the lower and upper hemispheres [15]. However, the tracker data shows that overall roll rotation was small, with the mean absolute roll 0.82˚ and the SD 0.51˚.

The high SD in the reversal errors shows that this metric is highly subject dependent. The SDs of reversal errors between model 'virtual listeners' were small compared to the behavioural results. This is not surprising, as the individual differences between subjects are likely not fully explained by the individual HRTFs. The same noise parameters were used for each individual, while it is likely that they differ per individual [57]. Furthermore, higher level processes such as listening strategies [58] or attention [59] will also be a cause for individual differences.

## Spatial analysis

Following the methods of visualisation of previous work, the localisation responses were modelled as elliptical Kent distributions [25, 60]. The centroids visualise the bias, i.e., the mean vector, of all responses for one source direction. The ellipsoid outlines visualise the equal probability contours of the distribution of responses. The major and minor axes of the ellipsoid are two SDs in length and represent the first two orthogonal 'principal components' of the dataset that account for the maximum amount of variance in the data. Fig 2 illustrates the Kent distribution of all responses for the frontal direction of condition BP.

Fig 3 visualises the behavioural and model localisation results per source direction on the sphere around the listener. Quadrant errors were excluded.

**Response bias.** The direction of the centroids of condition BP are similar to those found in [25]. More specifically, the centroids show a bias towards the audio-visual horizon and towards the interaural axis, i.e. the left and right ear. This supports the already strong evidence

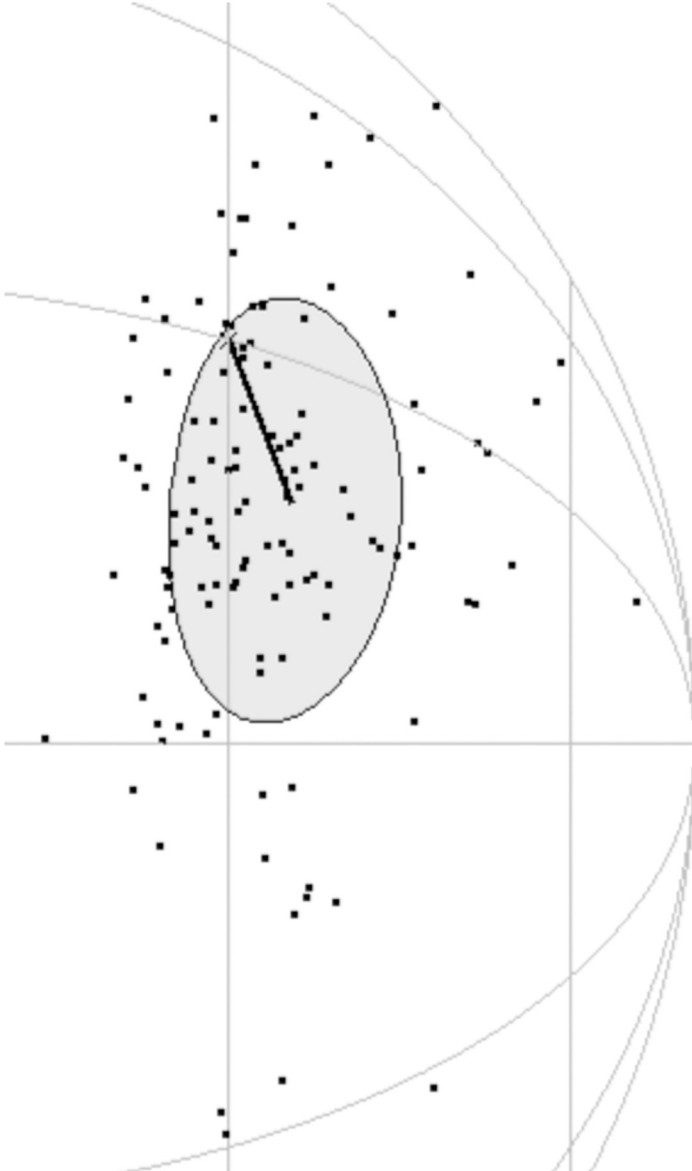

**Fig 2. Centroid and Kent distribution for condition BP and source direction (30°, 30°).** Black dots are the individual subject responses, from which the Kent distribution was calculated.

for a spatial prior on the horizon [24]. Several other studies have shown that human sound localisation displays a peripheral bias that increases with eccentricity [54, 61, 62].

In condition BA, the bias towards the horizon seemed to increase further. This can be explained if we assume an increase in uncertainty on the orientation of the head, or perhaps on the spectral cues during motion. Due to this increase in sensory noise, the relative strength of the spatial prior towards the horizon would increase. The biases did not change between conditions MP and MA. Possibly, this effect was not seen in the model because the head orientation was assumed to be perfectly known. Below we investigate the influence of increased sensorimotor noise.

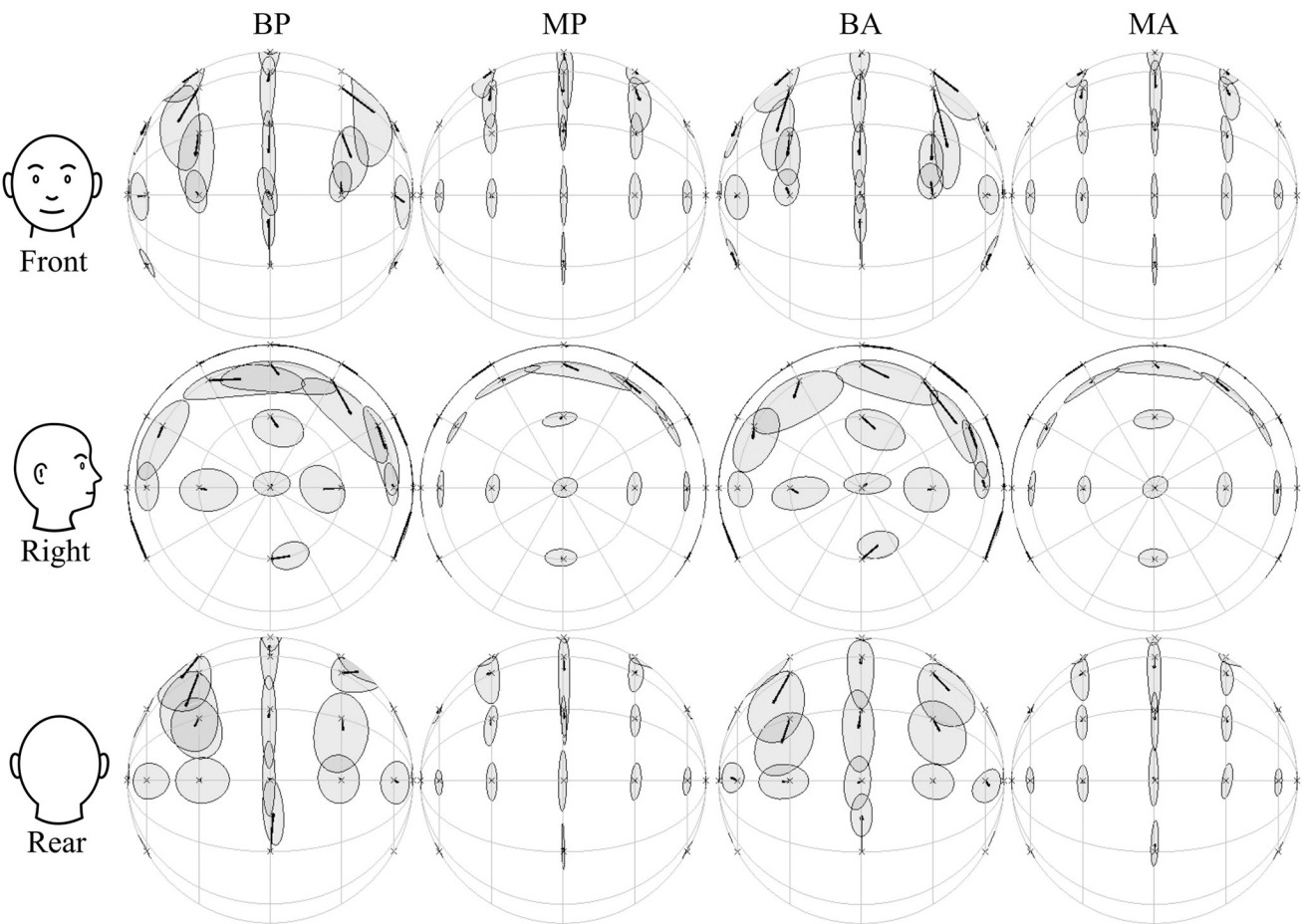

**Fig 3. Centroids and Kent distributions of behavioural (B) and modelled (M) responses in the passive (P) and active (A) conditions, averaged over eight subjects.** The rows show the same data viewed towards the front, the right, and the back of the head. Quadrant errors were excluded.

Two explanations for spatial biases have been proposed previously: 1) a Bayesian approach describes a pre-existing spatial prior for certain locations that is added to the sensory information, 2) alternatively, responses may be pulled to certain directions due to compressions and expansions in the sensory representations of auditory (and visual) space [62, 63]. Fig 3 shows that the selected spatial prior for the model results in vertical biases that are similar to the behavioural data. This is evidence that the spatial biases can, at least in part, be explained by a prior.

**Response variance.** Similar to the Kent distributions in [25], condition BA shows larger distributions for sources at higher elevations, and for sources in the rear hemisphere. The response spread is also highest along the polar dimension, more specifically, along the cones of confusion [64].

The spread in model responses also followed the cones of confusion. On the median plane the response distributions appear fairly similar. However, the response spread for sources at higher lateral angles was noticeably smaller than in the behavioural responses, especially for the lateral responses. This was to be expected from the lower local errors that were seen earlier.

The biggest difference was seen for sources in the rear. For conditions MP and MA, the response distributions for sources in the rear were nearly identical to those in front. On the contrary, the behavioural data contained a much larger spread in the rear, especially along the

lateral dimension. This suggests that the increase in variance, i.e., decrease in precision, found in the rear hemisphere of the behavioural results cannot be (fully) attributed to a lower spatial resolution in acoustic cues. Instead, a response or 'pointing' error may have been responsible. Even if the auditory system can perfectly estimate a source location, the action of pointing in a direction as a response may introduce an additional error. It is reasonable to hypothesise that sources behind the listener are more difficult to point towards consistently. Pointing errors have been modelled previously [20, 57], though only as a simple Gaussian noise source, which does not take into account the source direction. More research is required to accurately model the spatial dependence of a response error.

There were no large differences in Kent distributions between passive and active conditions, neither for the behavioural data nor for the model. One exception is the source position directly above the listener in the model predictions. In condition MP, a much higher polar spread is predicted than in condition BP. This can be explained by the Gaussian shape of the spatial prior, which affects sources at higher elevations more heavily than those around the horizon. Interestingly, this spread isn't visible in condition MA, which means that head rotation significantly improved estimation of this source direction and outweighed the spatial prior. This is another indicator of the available Wallach cue in the model simulations. This suggests that either perfect knowledge of the head orientation or lower noise on the ITD made head rotation more informative for the model than in the behavioural experiments.

**Quadrant errors.**   The spatial distribution of QEs was visualised in Fig 4. Condition BP reveals that QE rates were more common in the rear hemisphere than in the front: 59.4% and 40.6%, respectively. Most notable are the source directions directly in front of the listener, which showed nearly no QEs at all. Condition MP also shows more QEs in the back than in the front: 74.6% and 25.4%. This suggests that acoustic information accounts for most of the quadrant errors found. However, as was found in the Kent distributions, the source positions above the listener showed very different results between conditions BP and MP, this was caused by the spatial prior towards the horizon.

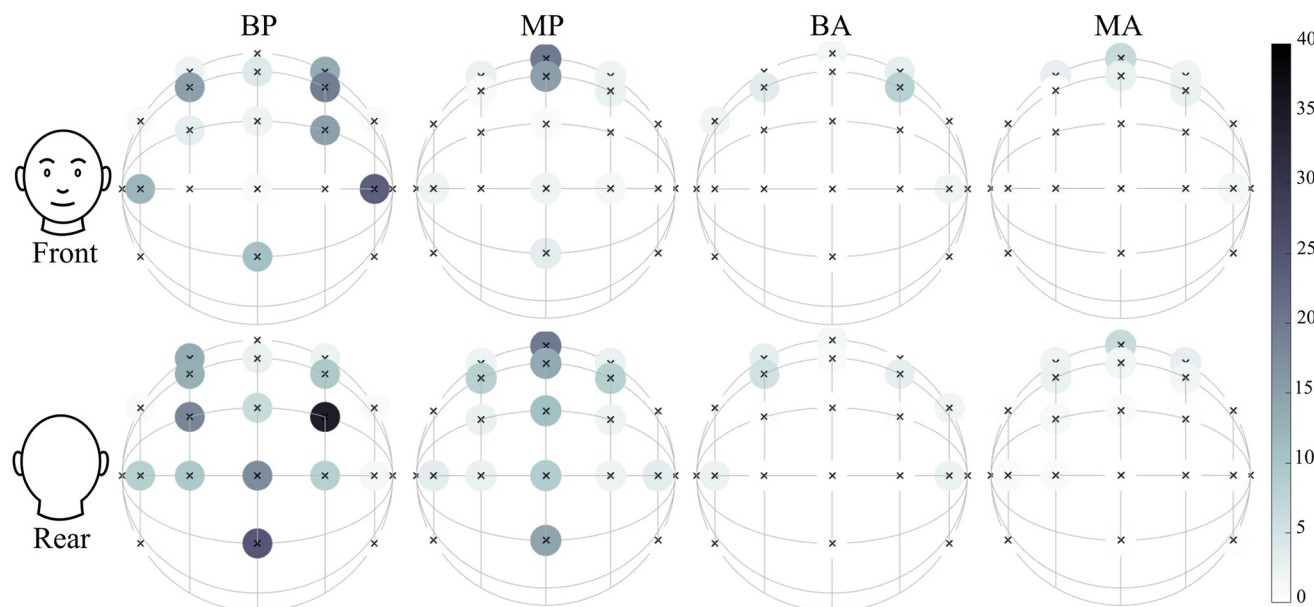

**Fig 4. Quadrant error rates [%] of behavioural (B) and modelled (M) responses in the passive (P) and active (A) conditions, averaged over eight subjects.** The rows show the same data viewed towards the front and the back of the head.

Previous studies have shown that the quantity and spatial distribution of reversal errors were highly subject-dependent. For example, one study found an even higher percentage of QEs to be located in the rear hemisphere than in the present study [53]. In another study, only two subjects showed a significant majority of reversal errors in the rear, while two showed a majority in the front, and two showed an equal distribution [56]. Similarly, this study contained three subjects with a majority in the back, one with a majority in the front, and four with no clear preference. For the localisation of low-pass stimuli, most confusions happened for sources in the front, note that this may be because sources from behind undergo more filtering than sources from the front [65].

From the present findings and the available literature, it is apparent that reversal errors involve a complex process that differs between individuals.

## Effect of the ITD noise

Table 1 shows that the lateral RMSE was too small compared to the behavioural results. Here, we tested new values of $\sigma_{itd}$ to investigate whether this parameter is the cause of the discrepancy. The results are plotted in Fig 5.

The lateral error increased monotonically with $\sigma_{itd}$. However, even for $\sigma_{itd} = 3°$, the errors were still lower than the behavioural results. This shows that the ITD noise alone cannot account for the discrepancy in lateral error. As expected, polar errors remained mostly unaffected in the passive condition, as the static ITD contains little to no information on the polar angle. However, the reduction in polar error in the active condition (due to the Wallach cue) was only visible for $\sigma_{itd} < 1.2°$. Minimising $\sigma_{itd}$ led to a complete removal of QEs in the active condition, whereas maximising it made the passive and active conditions near-identical. Together, the results suggest that a low noise on the ITD cue is essential to utilise the dynamic cue that resolves reversal errors when moving the head. They also show that the value of $\sigma_{itd} = 0.6$ that was derived from a previous experiment is a plausible value [21].

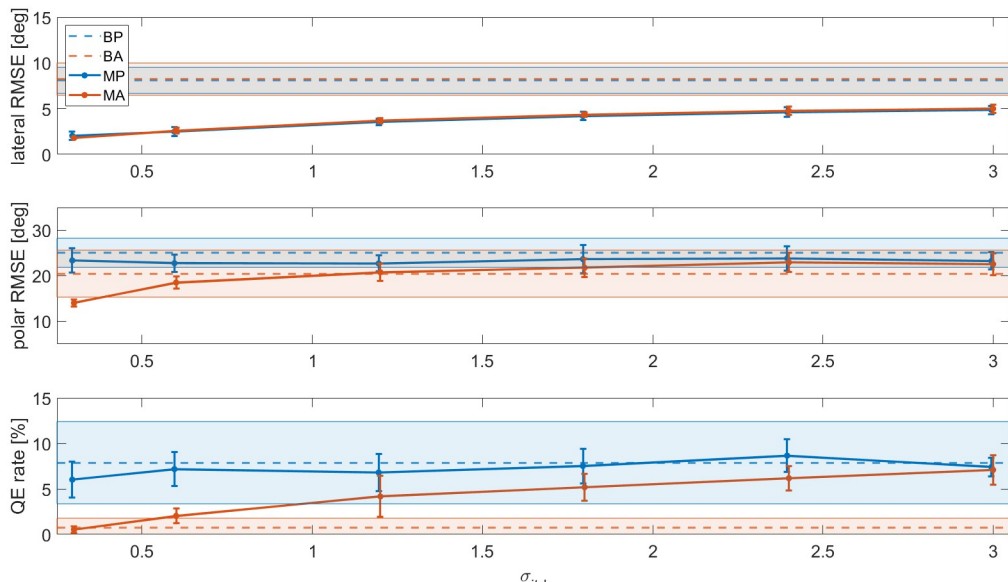

**Fig 5. Lateral RMSE, polar RMSE and QE rate of the modelled data as a function of $\sigma_{itd}$ (in units of the JND).** Blue markers are passive results, orange markers are active results. The markers and the error bars represent the mean and standard deviation over the eight modelled subjects. For reference, the dashed lines and the coloured areas show the behavioural means and standard deviations over the eight subjects, respectively.

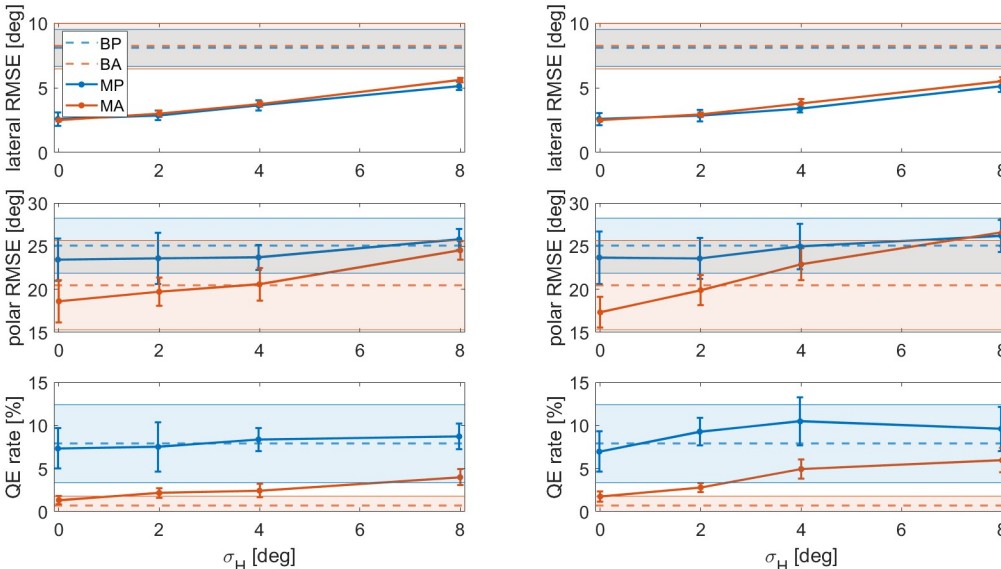

**Fig 6. Lateral RMSE, polar RMSE and QE rate of the modelled data as a function head orientation measurement noise $\sigma_H$, with head control noise $\sigma_u = 0°$ (left column) and $\sigma_u = 8°$.** Blue markers are passive results, orange markers are active results. The markers and the error bars represent the mean and standard deviation over the eight modelled subjects. For reference, the dashed lines and the coloured areas show the behavioural means and standard deviations over the eight subjects, respectively.

## Effect of the sensorimotor noise

For the initial simulations, the head orientation was assumed to be perfectly known. Here we investigated the effects of an increased uncertainty on the head orientation. The simulations were rerun with different values for $\sigma_H$ and $\sigma_u$. The results are plotted in Fig 6.

The results show that a larger value for $\sigma_H$ increased the lateral error, similar to $\sigma_{itd}$. There appeared to be no interaction effect with $\sigma_u$.

Polar error monotonically increased with $\sigma_H$, though the active condition seemed to suffer slightly more. In other words, the difference in polar error between condition MP and MA became smaller as $\sigma_H$ increased. This effect was even stronger with a high $\sigma_u$. This suggests that uncertainty on the head position is the reason why the Wallach cue cannot be used by human listeners.

For $\sigma_u = 0°$, the QE rate was affected slightly for higher values of $\sigma_H$, notably less than when $\sigma_{itd}$ was increased. For $\sigma_u = 8°$, the effects were larger, and even condition MP seemed to suffer more QEs.

Together, this leads to the conclusion that the uncertainty on the sensorimotor measurement of the head orientation, $\sigma_H$, can account for several (though not all) of the differences found between the behavioural data and the model output, but that the noise on the execution of the control signal, $\sigma_u$ needs to be low.

Note that $\sigma_H$ and $\sigma_u$ will likely be higher in the active condition than in the passive condition, as motor noise is multiplicative [42].

## Effect of the spatial prior

The bias vectors in the behavioural results appeared slightly stronger than for the simulations with $\sigma_p = 30$. This implies that a stronger spatial prior may be necessary. There are many different possible spatial prior shapes, e.g. a Laplace distribution [66], or prioritising the front or

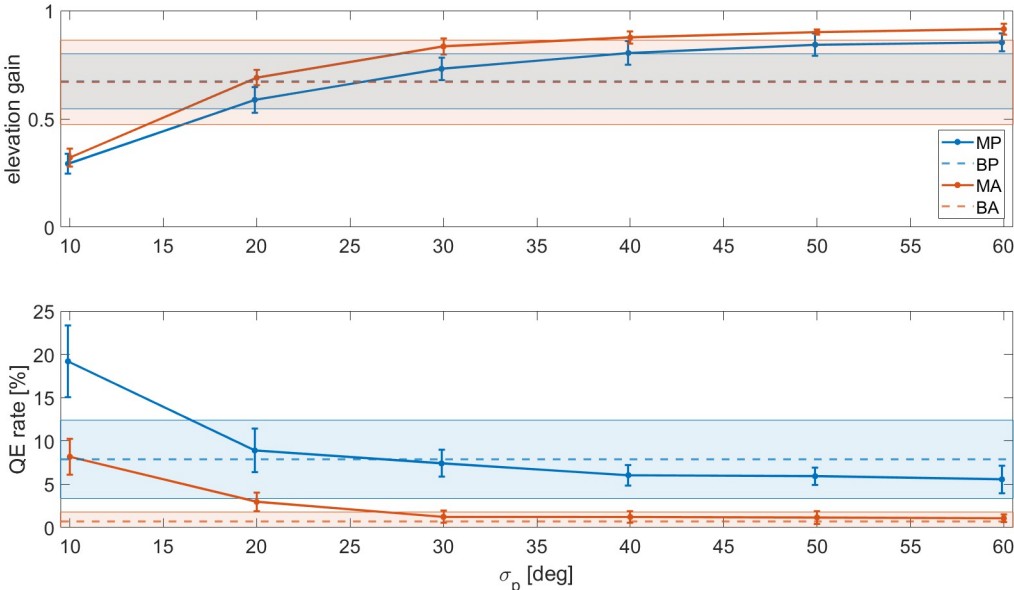

**Fig 7. Elevation gain [24] and QE rate of the modelled data as a function of $\sigma_p$ (in degrees).** Blue markers are passive results, orange markers are dynamic results. The markers and the error bars represent the mean and standard deviation over the eight modelled subjects. For reference, the dashed lines and the coloured areas show the behavioural means and standard deviations over the eight subjects, respectively.

high lateral angles [67]. In this study, the analysis was restricted to the horizontal prior. Fig 7 shows the elevation gain [24] (i.e., the slope of a linear regression between responses and true directions) and QE rate for different values of $\sigma_p$. As we are interested in the 'pull' towards the horizon, the elevation gain is a more appropriate indicator than the polar RMSE.

The plot suggests that the correct prior $\sigma_p$ lies between 20˚ and 30˚, where the gain and the QE rates are closest to the behavioural results. Thus, $\sigma_p = 30$ was again a fairly good estimate, though the large standard deviations between subjects in the behavioural data suggest that the strength of the prior may be subject-dependent. As $\sigma_p$ increases, the prior approaches a uniform distribution and the relative weight of the sensory information will increase. The plot shows that, as a result, the responses approach an elevation gain of 1. This suggests that the bias present in human responses is not due to acoustic factors, but indeed due to a prior towards the horizon.

However, there remains a problem in the distribution of the errors. First, it was noted that the prior affects sources at higher elevations more heavily than those around the horizon, resulting in an unrealistically high spread in responses and high QE rate for sources above the listener (see Figs 3 and 4, condition MP). There also remains the lateral bias that is unaccounted for with the spatial prior tested in this study. These discrepancies could mean that humans have an additional auditory spatial bias upwards and towards higher lateral angles. Alternatively, as stated earlier, it is possible that responses are pulled or 'snap' to certain directions due to the sensory representations of space [62, 67]. The latter explanation would be an example of non-ideal observer behaviour.

## Effect of the time step

To investigate the effect of the time step size $\Delta t$, the model was rerun with different update rates. Note that each simulated trial always contained at least two time steps at the start and at

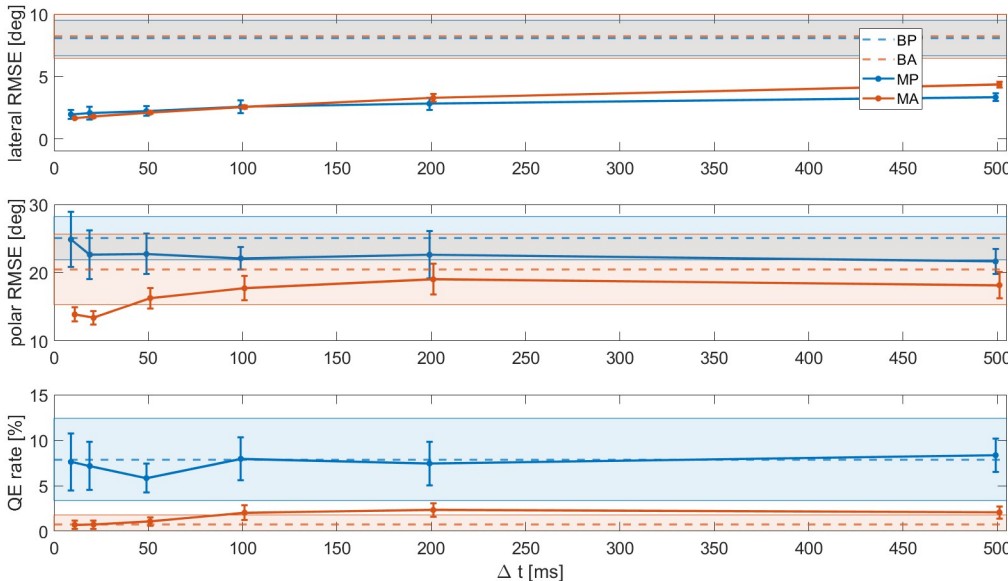

**Fig 8. Lateral RMSE, polar RMSE, and QE rates of the modelled data as a function of time step size Δ*t*.** The symbols show the averages and the error bars represent ±1 SDs over the (virtual) subjects. For reference, the horizontal dashed lines show the behavioural data.

the end of the stimulus duration, to make dynamic cues available. The results are shown in Fig 8.

Generally, it seems that a smaller Δ*t* makes localisation performance slightly more accurate, this is the result of more looks being available. This improvement is mostly visible in the polar RMSE of condition MA, which shows that the benefit of the Wallach cue becomes more prevalent if more ITD looks are allowed. Unsurprisingly, a similar effect was seen when the noise on the ITD looks was kept low. For the other metrics, the improvement performance is surprisingly small. Regarding the QE rate, it appears that two ITD looks (one at the start and one at the end of rotation) were sufficient to prevent most errors, and that any look in between is somewhat redundant. Finally, there appeared an unexpected dip in the QE rate of condition MP at $\Delta_t = 50ms$. A repetition of the model simulations revealed that this was a statistical anomaly.

## Conclusions

This article introduced a Bayesian ideal observer model that enables a bottom-up investigation of human performance in the task of active sound localisation. In order to investigate to what extent humans perform as ideal observers, the model output was compared to behavioral results obtained in a free-field localisation experiment.

With parameters selected a priori, i.e., without the use of any post-hoc fit to the behavioral data, the model predicted and explained the human performance in a general sense. This is an encouraging finding, supporting the hypothesis that model parameters can be derived a-priori based on general behavioural experiments. Furthermore, as the model reproduced human performance while processing changes only in ITD, it also confirms the earlier finding that humans do not utilise dynamic spectral cues for localisation, at least during small head rotations [6].

In a more detailed spatial analysis, the model predictions deviated from the behavioral data. The largest differences were found for sources to the rear and above the listener. We

investigated in detail the conditions where the model agreed or deviated from the behavioural data by studying the effects of several model parameters on the predictions. The ITD noise parameter revealed that the ITD cue alone does not account for the lateral RMSE and QE rate found in behavioural data. The uncertainty on the head orientation had a significant effect and was able to partially explain the behavioural data.

The discrepancies we found between the model predictions and behavioural data are important for future investigations. First, there is a high variance in behavioral responses for sources behind the listener, whereas the model showed little difference between the front and rear hemisphere. This indicates that these errors result from non-acoustic factors, such as the pointing error. Second, human listeners showed a response bias towards larger lateral angles, which was not seen in the model predictions. The origins of this discrepancy remain an open question, with a lateral spatial prior or a stretched sensory representation of space as potential candidates. Third, the model predictions showed a higher response variance and QE rates for sources placed above the listener. In the model, this might be an effect of the Gaussian spatial prior towards the horizon, which might not fully reflect the spatial prior of the human listeners, or even point to a more complex mechanisms at play.

Our framework can be applied in the future to a variety of phenomena that have been identified in previous studies on active sound localisation, such as the improvement of elevation perception with yaw movements for low-pass stimuli [3], elimination of FBCs for low-pass stimuli [37], and the relative weight of dynamic ILD and ITD in the localisation process [68].

## Acknowledgments

The authors would like to thank Philip Leong for providing the MATLAB code to the 'Spak' spherical data processing tool. Part of this code was rewritten for the purpose of this work and integrated into the AMT.

## Author Contributions

**Conceptualization:** Glen McLachlan, Herbert Peremans.

**Data curation:** Glen McLachlan.

**Formal analysis:** Glen McLachlan.

**Funding acquisition:** Herbert Peremans.

**Investigation:** Glen McLachlan.

**Methodology:** Glen McLachlan, Michael Mihocic.

**Software:** Glen McLachlan, Michael Mihocic.

**Supervision:** Piotr Majdak, Herbert Peremans.

**Validation:** Glen McLachlan.

**Visualization:** Glen McLachlan.

**Writing – original draft:** Glen McLachlan.

**Writing – review & editing:** Glen McLachlan, Piotr Majdak, Jonas Reijniers, Herbert Peremans.

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
