## [Decision Letter · Decision Letter 0]

27 Jul 2024

Dear Mr McLachlan,

Thank you very much for submitting your manuscript "Insights into dynamic sound localisation: A direction-dependent comparison between human listeners and a Bayesian model." for consideration at PLOS Computational Biology.

As with all papers reviewed by the journal, your manuscript was reviewed by members of the editorial board and by several independent reviewers. In light of the reviews (below this email), we would like to invite the resubmission of a significantly-revised version that takes into account the reviewers' comments.

We cannot make any decision about publication until we have seen the revised manuscript and your response to the reviewers' comments. Your revised manuscript is also likely to be sent to reviewers for further evaluation.

Sincerely,

Elia Formisano, Ph.D.

Guest Editor

PLOS Computational Biology

Andrea E. Martin

Section Editor

PLOS Computational Biology

Reviewer's Responses to Questions

**Comments to the Authors:**

Reviewer #1: Yes, I will upload my review as a word file

Reviewer #2: General issues:

1) A qualitative description of how the model does and does not operate would be helpful especially for readers unfamiliar with the authors' previous paper introducing the model. By this I mean in particular that it should be made explicit that the model uses motion implicitly and does not actually calculate any dynamic cues (such as the sign or magnitude of the ratio of delta_ITD to delta_yaw). The templates contain only static cues associated with each location in head-centred coordinates, and (as I understand it) motion essentially serves to eliminate static ambiguities (e.g. ITD-based cone-of-confusion and possible front/back spectral cue similarities) by continually re-registering the head-centred coordinates with world-centred (or torso- or zero-yaw-centred) ones to allow template similarity to "pile up" near the veridical source location over multiple temporal snapshots. Additionally, it would be helpful to add a figure showing how the posterior probability disribution develops over the course of a simulated head-turn, eliminating erroneous location modes in the PDF.

2) Clarification of the relationship between the conditions of simulation and the behavioural experiment is needed. This caused me considerable confusion on first reading because I was expecting more congruency between the conditions. In the simulation, the head executes a yaw starting at rest at 0 degrees with an acceleration of 20 deg/s/s. The simulation is then run over varying durations (5-1000 ms) of that motion, which results in different amounts of total yaw, depending on duration (0.1 degrees for 100 ms, 2.5 degrees for 500 ms, 10 degrees for 1000 ms), although this is not emphasized in the text. In contrast, in the behavioural experiment, duration is not varied, but rather the participant initiates a head turn after the onset of the stimulus, and the stimulus is gated off after the execution of 10 degrees of yaw rotation. So here duration may vary but total yaw does not. The statistics of the resulting behavioral head motions (latencies, accelerations, and durations) and how well the stimulated motion aligns with them is not discussed, but should be. This would inform the authors' choice of which simulated durations to concentrate on. Given the 20 deg/s/s acceleration, the full 1000 ms would be required for 10 degrees of rotation.

3) Given the closing paragraph ("The present analysis shows that this model can serve as a powerful framework ...") I suggest adding some discussion of the possible application of the model to a variety of different phenomena that have arisen in previous and recent investigations of dynamic localization and of how it might need to be modified (in terms of: explicit weighting of binaural versus spectral cues; incorporating ILD or explicit dynamic cues; relaxing the assumption of a stationary source) to account for:

- substantial elimination of F/B errors with low-pass stimuli (no high-frequency spectral cues) and over ~100-ms constant duration regardless of yaw velocity

- authors' previous report (Front. Neurosci. 2023) that dynamic spectral cues are actually not important for small head turns

- Perrott & Noble's 1997 report of low-pass elevation perception with yaw movements

- Wallach illusion succeeding for low-pass stimuli but not for ones with potent real (e.g. wideband noise) or phantom (e.g. high-frequency narrow-band noise) spectral cues. Apparently the assumption of stationarity is abandoned for the latter stimulus types.

Line-by-line:

51-52: "The proposed model utilises a ‘template-matching’ procedure which requires an acoustic template TA that the observed information is compared to.": Although the authors' model uses more sophisticated statistical techniques, a reference to Middlebrooks 1992 (Narrow‐band sound localization related to external ear acoustics. JASA, 92(5), 2607-2624.) would be appropriate as an originator of such "similarity over the sphere" -type models.

52: "an acoustic template T_A": wouldn't it be clearer to refer to this as a _set_ of templates, one for each location in head-centred coordinates?

58-64: It makes sense to use only ITD, and not ILD, since ITD appears to dominate lateral-angle perception for such signals anechoically, but it would be good to mention this model design decision, and that fact that ILD would provide some source-spectrum independence that the monaural spectra lack.

68: "bandwidths (ERBs) following": missing text after "following"?

71: "This stage simulates a simplified processing of the inner hair cell [20]": although the authors of [20] may also have used such an IHC model, this does not seem like the best reference for this approach.

72: "filterbank outputs were then summed over time": Not clear what this means.

74: "the ITD and the monaural spectral vectors for both ears are combined": this gives one vector element for ITD, but 64 elements for spectral information; how does this affect the relative influence of ITD and spectral cues?

144: "marginalisation over all possible head orientations": does this literally mean _all_ possible orientations, or over the probability distribution of head orientations based on the accumulated sensorimotor evidence?

193: "Because this model computes the posterior recursively, as opposed to a single computation": is this really 'recursive' (which to me connotes a process that calls itself repeatedly and then "backs out" again), or merely "iterative" or cumulative? I would not call an IIR filter "recursive” just because past outputs are current inputs.

226-227: "A head-mounted display ... was used for the visual presentation of virtual reality (VR)": Describe the purpose of the VR. Were listeners' HRTFs measured while wearing the HMD, or can the authors comment on its acoustical or behavioural effect?

244-245: "Each trial used the same noise realisation, thus there was no variation of the spectral content between trials.": Not strictly true since each trial would involve a different duration cut-off of the noise waveform.

272-273, Table 2: Mention in the caption that data from other studies are included here. Define "B-Static", "M-Dynamic", etc. Why is B-Dynamic listed as 500 ms? Participants would have had a distribution of distributions, so is this representative?

320: "The model localisation errors are reported for 100 ms. For this duration, the mean errors were similar to that of the behavioural data.": Provide more rationale for the choice of this comparison. For example, why would or would not a duration matching the listeners' mean duration be more apt? How best to compare durations between simulation and behavioural data when there must have been a considerable latency for listeners' initiation of movement?

Fig 1, Fig 2, Fig 7: Specify the model duration in the captions and/or column labels.

359-360: "responses may be pulled to certain directions due to compressions and expansions in the sensory representations of auditory (and visual) space [51]": A reference to Brimijoin 2018 (Angle-dependent distortions in the perceptual topology of acoustic space. Trends in Hearing, 22) would be appropriate here.

367-368: "The biases did not change in the dynamic simulations, whereas the biases increased in the behavioural results.": Clarify wording to emphasize that the "change" or "increase" is for the dynamic situation relative to the static one.

411-412: "However, previous studies have shown that the quantity and spatial distribution of reversal errors were highly subject-dependent.": Might it be good also to look at localization of low-pass stimuli (lacking high-frequency spectral cues) to inform the idea of a front/back spatial prior? I recall a paper by Simon Carlile showing a roughly 1/3 split between individuals consistently localizing such stimuli to the front, to the rear, or evenly mixed.

426-427: "The model results revealed that stimulus duration was a significant factor in the localisation task": Here it should be made explicit that duration is confounded with amount of rotation.

436-438: "When head movements are allowed, a stimulus duration of approximately 100 ms seems to be required to provide a substantial benefit from the head movement [59]": Note that in that study, the 100-ms "threshold" applied to: stimuli lacking high-frequency spectral cues, widely different amounts of rotation depending on head-turn velocity, and to roughly constant velocity over the 1oo-ms period (i.e. not starting from rest).

439: "Fig 3 shows that the model reached a performance plateau around 500 ms": Point out that it is much earlier than 500 ms for QE rate, also mention how much rotation 500 ms corresponds to.

448-449: "Alternatively, it may be that humans only optimally integrate acoustic information up to 100 ms of listening": Adjust wording to make clear this is about localization; temporal integration for detection and loudness is 200-300 ms.

454-455: "Assuming a possibly moving sound source would put a premium on forgetting old acoustic information as the state of the world would have changed in the meantime": This is also an argument for adding explicit, instantaneous dynamic cues to the model.

456-457: "the lateral RMSE best matched the behavioural results at a single time step of 5 ms (5.5◦)." Were other time steps tried; how is it known that 5 ms was "best? Also, how is 5 ms equated with "5.5 degrees"?

457-458: "This suggests that the ITD information may not be integrated at all and is instead remeasured and reevaluated entirely at each time step." Am not following the argument here ... unpack?

**Have the authors made all data and (if applicable) computational code underlying the findings in their manuscript fully available?**

Reviewer #1: Yes

Reviewer #2: Yes

PLOS authors have the option to publish the peer review history of their article (what does this mean?). If published, this will include your full peer review and any attached files.

Reviewer #1: **Yes: **Snandan Sharma

Reviewer #2: No
---

## [Decision Letter · Decision Letter 1]

9 Dec 2024

PCOMPBIOL-D-24-00691R1

Bayesian active sound localisation: to what extent do humans

perform like an ideal-observer?

PLOS Computational Biology

Dear Dr. McLachlan,

Thank you for submitting your manuscript to PLOS Computational Biology. After careful consideration, we are quite likely to move forward with publication of the manuscript if the final minor comments of the Reviewer are addressed.  Therefore, we invite you to submit a revised version of the manuscript that addresses the points raised during the review process. 

Please submit your revised manuscript within 30 days Feb 08 2025 11:59PM. If you will need more time than this to complete your revisions, please reply to this message or contact the journal office at ploscompbiol@plos.org. Please include the following items when submitting your revised manuscript:

We look forward to receiving your revised manuscript.

Kind regards,

Andrea E. Martin, Ph.D.

Section Editor

PLOS Computational Biology

Feilim Mac Gabhann

Editor-in-Chief

PLOS Computational Biology

Jason Papin

Editor-in-Chief

PLOS Computational Biology

**Journal Requirements:**

1 Thank you for including an Ethics Statement for your study. Please include:

i) The approval number(s), or a statement that approval was granted by the named board(s).

2) Please upload the figures in a correct numerical order in the submission form.

**Comments to the Authors:**

**Please note that the review is uploaded as an attachment.**

**Reviewers' comments:**

Reviewer's Responses to Questions

Reviewer #2: The revised manuscript as a whole, the new approach to simulation (matching participants' motion trial-by-trial), and the re-specification of the goal of the study are so much easier to follow than the original version. Very straightforward. My only comments relate to one typo and three minor wording suggestions.

252: "this decision was made as it was a higher precision error was found" - delete "it was"

458 & 473: > "The results are plotted"

541-544: Needs a little more 'logic glue'. Suggest: "Furthermore, as the model reproduced human performance while processing changes only in ITD, it also confirms the earlier finding that humans do not utilise dynamic spectral cues for localisation, at least during small head rotations"

**Have the authors made all data and (if applicable) computational code underlying the findings in their manuscript fully available?**

Reviewer #2: Yes

PLOS authors have the option to publish the peer review history of their article (what does this mean?). If published, this will include your full peer review and any attached files.

Reviewer #2: No

**Figure resubmission:**
---

## [Editor Report · Decision Letter 2]

16 Dec 2024

Dear Mr McLachlan,

We are pleased to inform you that your manuscript 'Bayesian active sound localisation: to what extent do humans perform like an ideal-observer?' has been provisionally accepted for publication in PLOS Computational Biology.

Best regards,

Andrea E. Martin, Ph.D.

Section Editor

PLOS Computational Biology

---

## [Editor Report · Acceptance letter]

31 Dec 2024

PCOMPBIOL-D-24-00691R2 

Bayesian active sound localisation: to what extent do humans perform like an ideal-observer?

Dear Dr McLachlan,

I am pleased to inform you that your manuscript has been formally accepted for publication in PLOS Computational Biology. Your manuscript is now with our production department and you will be notified of the publication date in due course.

With kind regards,

Lilla Horvath
